# Statement: Relevance to the Track

October 2023

The work presented in this paper is highly relevant to the track "Graph Algorithms and Modeling for the Web" as it directly addresses a significant Web-related research challenge, under-represented or ambiguous regions in the relational data. Our research focuses on advancing the capabilities of Graph Neural Networks (GNNs) indealing with underrepresented regions, such as ambiguous nodes and regions that exhibit irregular homophily/heterophily or neighborhood patterns.

The Web produces data of diverse and intricate graph structures, characterized by diverse connectivity patterns, varying class distributions, and ambiguous regions, making the efficient analysis and representation learning of web data an ongoing challenge. Our work investigate the ambiguity problem within GNN-produced node embeddings when applied to web-related graph data. Concretely, we utilize the given relation information and prediction distributions to identify ambiguous regions, and provide richer learning signals with a neighborhood contrast algorithm.

In conclusion, our research directly aligns with the goals of the "Graph Algorithms and Modeling for the Web" track, and our method is designed fully exploiting the opportunity of relational data. In this work, we provide practical solutions for enhancing the utility of GNNs in learning high-quality representations across regions of the graph.

# Disambiguated Node Classification with Graph Neural Networks

Anonymous Author(s)

## ABSTRACT

Graph Neural Networks (GNNs) have demonstrated significant success in learning from graph-structured data across various domains. Despite their great successful, one critical challenge is often overlooked by existing works, i.e., the learning of message propagation that can generalize effectively to underrepresented graph regions. These minority regions often exhibit irregular homophily/heterophily patterns and diverse neighborhood class distributions, resulting in ambiguity. In this work, we investigate the ambiguity problem within GNNs, its impact on representation learning, and the development of richer supervision signals to fight against this problem. We conduct a fine-grained evaluation of GNN, analyzing the existence of ambiguity in different graph regions and its relation with node positions. To disambiguate node embeddings, we propose a novel method, DisamGCL, which exploits additional optimization guidance to enhance representation learning, particularly for nodes in ambiguous regions. DisamGCL identifies ambiguous nodes based on temporal inconsistency of predictions and introduces a disambiguation regularization by employing contrastive learning in a topology-aware manner. DisamGCL promotes discriminativity of node representations and can alleviating semantic mixing caused by message propagation, effectively addressing the ambiguity problem. Empirical results validate the efficiency of DisamGCL and highlight its potential to improve GNN performance in underrepresented graph regions.

## ACM Reference Format:

Anonymous Author(s). 2023. Disambiguated Node Classification with Graph Neural Networks. In *Proceedings of ACM Conference (Conference'17)*. ACM, New York, NY, USA, 10 pages. https://doi.org/10.1145/nnnnnnn.nnnnnnn

## 1 INTRODUCTION

In recent years, learning from graph-structured data has received significant attention due to its prevalence in various domains [7, 13, 64], including social networks, molecular structures and knowledge graphs. These applications necessitate the utilization of rich relational information among entities. Graph neural networks (GNNs) [50] offers a powerful framework to combine graph signal processing and convolutions and have shown great ability in representation learning on graphs. Various GNNs have been proposed. Most of them adopt message-passing process which learns a node representation by iteratively aggregating its neighbors' representations [14, 15, 24].

Along with their success, several concerns have aroused regarding GNNs' potential weaknesses, e.g., GNNs may show weak performance occasionally and may even be outperformed by simple neural networks that do not leverage relational information at all [30, 34]. Recent studies suggest that the root cause of this phenomenon may be the inductive bias inherent in the message propagation process [32, 54]. For example, a number of works [3, 62, 66] observe that when the *heterophily* level of a graph is high, i.e., when connected nodes are more likely to belong to different classes or posses different attributes, the performance of GNNs tends to decline significantly. To address this issue, many studies have proposed revisions to the message propagation process, with strategies such as edge refinement [1, 34], high-pass signal filters [3, 18], etc.

Despite the ongoing efforts to design more expressive GNN architectures, there is a crucial problem that has often been overlooked: real-world graphs can exhibit diverse regions with varying degrees of heterophily and neighborhood patterns. In such cases, GNN models may struggle to perform well in regions with irregular or infrequent structures. Some motivating examples of such under-represented regions are provided in Fig. 1. These regions could include homophilous regions in a heterophily graph or nodes adjacent to both minority and majority classes simultaneously. In semi-supervised node classification, the most common graph learning task, only a small fraction of nodes from each class are available for training. The message propagation mechanism is learned based on these few-shot labeled nodes and is expected to generalize well across the entire graph. Considering the inductive bias introduced by message-passing, we argue that such graph regions may exhibit distinct neighborhood patterns that are underrepresented, which we term as "ambiguous regions" or "mixed regions". The learning of message-passing-based GNNs is dominated by majority nodes, potentially resulting in ambiguous and indistinguishable representations for ambiguous regions despite whether the whole graph is heterophilous or not. Modern GNNs may have sufficient expressive power in distinguishing nodes [46] yet the insufficient supervision signals could be the problem in learning general and robust models [17].

In light of these challenges, our work focuses on providing richer optimization guidance to GNNs for effectively learning nodes in these ambiguous regions. This ambiguity problem in the representation space would be further complicated by the semi-supervised nodes and potential class imbalances during the learning process [5, 61]. To validate this insight, we empirically analyze the confusion of several representative GNNs on different regions of real-world graph datasets. We divide the graph into multiple groups based on class size and neighborhood distributions, and analyze the model performance across these regions. Empirical results in Sec. 4 reveal that ambiguity exists in different regions and is influenced by imbalanced classes.

Therefore, in this paper, we propose to enhance representation learning for classifying nodes in ambiguous regions by exploiting additional optimization guidance. The primary challenges are

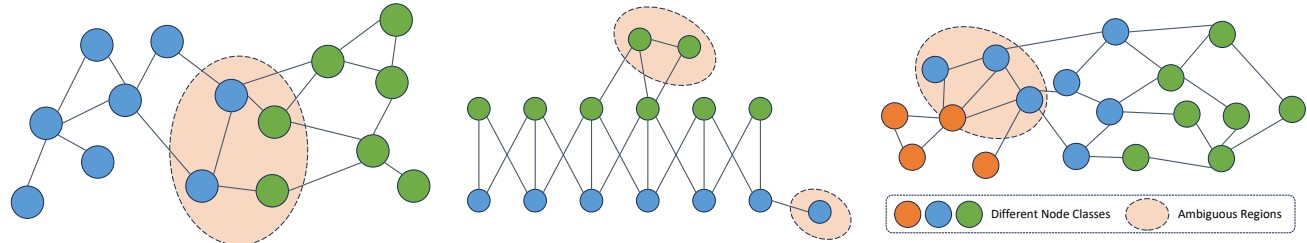

(a) Heterophily regions in a homophily graph    (b) Homophily regions in a heterophily graph    (c) Regions neighboring minority classes

Figure 1: Examples of ambiguous regions, which exhibit under-represented local structure patterns.

twofold: (1) *How can we identify these ambiguous nodes given limited label information?* (2) *How can we guide the GNN to address the nodes that the model is confused about?* To tackle these challenges, we introduce a novel method, DisamGCL. DisamGCL identifies ambiguous nodes based on the temporal inconsistency of prediction outputs, as underrepresented instances tend to exhibit more unstable behaviors alongside the variation of knowledge captured by GNN during training. We then design a disambiguation regularization objective to reduce noisy messages and enhance discrimination in a topology-aware manner by encouraging disparity between target nodes and their direct neighbors with different semantics. A contrastive learning framework is adopted, incorporating dissimilar neighbors as negative samples to promote discriminativity and prevent semantic mixing, thereby alleviating the ambiguity problem. Our **main contributions** are:

- We investigate a novel problem: the performance degradation observed in underrepresented graph regions with distinct ego-graph distributions, stemming from the inherent inductive bias of message propagation in GNNs.
- We conduct a fine-grained evaluation of representative GNN variants to assess the ambiguity problem. This analysis underscores the issue of under-representation during training, shedding light on the challenges associated with ambiguous regions.
- We propose a novel framework, DisamGCL, which can automatically identify ambiguous nodes and dynamically augment the learning objective with a contrastive learning framework. Empirical results show the effectiveness of the proposed DisamGCL

## 2 RELATED WORK

### 2.1 Graph Neural Networks

With the growing demand for learning on relational data structures [9, 13], various graph neural network (GNN) architectures have emerged, encompassing designs rooted in convolutional neural networks [4, 25], recurrent neural networks [27, 40], and transformers [12]. Despite their architectural diversity, the majority of GNNs operate within the paradigm of message-passing [14], iteratively updating nodes by aggregating messages from their local neighborhoods. For example, GCN [25] conveys messages from neighboring nodes with fixed weights, while GAT [44] employs self-attention mechanisms to learn varying attention scores for dynamic message selection. Noteworthy extensions to traditional GNNs include approaches such as Prototypical-GNN [28] and Memory-Augmented GNNs [2, 52], which introduce explicit prototypes to

hierarchically model motif structures and enhance data efficiency. Additionally, works like Factorizable Graphs [56] and Decoupled Graphs [51] propose methods to unveil latent groups of nodes or edges and convey messages on disentangled graphs. Recent investigations have also delved into the trustworthiness [11, 36, 37, 48] and interpretability [10, 58] of GNNs.

### 2.2 Contrastive Learning in GNNs

Recently, remarkable progress has been made to adapt Contrastive Learning (CL) techniques [6, 26] for the graph domain, by constructing multiple graph views via augmentations and maximizing the mutual information between instances with similar semantics (positive samples) [31, 69]. Existing methods mainly differ in the selection of graph augmentation techniques and contrastive pretext tasks. Popular augmentations include node-level attribute manipulation [20, 33, 45], topology-level edge modification [57, 67], graph diffusions [16, 22] to connect nodes with indirectly connected neighbors, etc. Different pretext tasks can be constructed based on the assumption of similar "semantics". Mainstream strategies include same-scale contrasts and cross-scale contrasts. In same-scale contrasts, positive samples are often chosen as the congruent node representations in other view while representation of other nodes are used as negative samples [57, 70]. In cross-scale contrasts, MI is maximized between global graph embeddings and local sub-graph representations or node representations of the same view [33, 45]. Recently, one critical issue that has gained research attention is the problem of graph heterophily. Graphs in real-world applications often exhibit heterophily, where nodes within the same community or group tend to connect with nodes from different communities, reflecting diverse relationships or interactions. This phenomenon poses a unique challenge for traditional graph neural networks (GNNs) that are typically designed to work under the assumption of homophily, where nodes within the same community preferentially connect with each other.

### 2.3 Heterophily in GNNs

Graphs in real-world applications often exhibit heterophily, where a node tend to connect to nodes of different classes or features, reflecting diverse relationships or interactions. Heterophily has received lots of attention in recent years [55, 63], and existing heterophilic GNNs are generally designed from two perspectives: (1) adding new nodes to the neighborhood to augment the propagated message; (2) flexible aggregation of neighborhood messages. Multi-hop

neighbors are explored in [1, 21, 47, 66] to augment the propagated messages, which tend to be more robust than using one-hop neighbors alone. Geom-GCN [34] and NL-GNN [30] discover potential neighbors by measuring the geometric relationships and representation similarities between node pairs respectively. Addressing the encoding of low-frequency and high-frequency graph signals, more powerful spectral kernels have been designed for dynamic aggregation [3, 18]. GPR-GNN [8] assigns learnable weights to combine the representations via the Generalized PageRank (GPR) technique. ACM [32] adaptively exploit beneficial neighbor information from different filter channels for each node.

It is important to note that while our research aligns with this direction, we primarily focus on addressing the disambiguation problem encountered by GNNs due to less common neighborhood patterns. Ambiguity in our context is determined not solely by comparing local and global homophily ratios but also by considering factors such as majority/minority class distinctions and node positions, as discussed in Fig. 2. This highlights a key distinction between our work and the aforementioned approaches

## 3 PRELIMINARY

### 3.1 Notations and Problem Definition

In this paper, we use $\mathbf{G} = (\mathbb{V}, \mathcal{E}; \mathbf{F}, \mathbf{A})$ to denote a graph, where $\mathbb{V}$ is the node set and $\mathcal{E} \subset \mathbb{V} \times \mathbb{V}$ is the edge set. Nodes are accompanied by an attribute matrix $\mathbf{F} \in \mathbb{R}^{|\mathbb{V}| \times d}$, and $i$-th row of $\mathbf{F}$ is the $d$-dimensional attributes of the corresponding node $i$. $\mathcal{E}$ is described by an adjacency matrix $\mathbf{A} \in \mathbb{R}^{|\mathbb{V}| \times |\mathbb{V}|}$. $A_{vu} = 1$ if there is an edge between node $v$ and $u$; otherwise, $A_{vu} = 0$. $\mathbf{Y} \in \mathbb{R}^{|\mathbb{V}|}$ is the class information for nodes in $\mathbf{G}$, obtained with an unknown labeling function, and $R(\mathbf{Y})$ is the number of classes.

We focus on the node-level classification task in this work. During training, $\mathbb{V}^L \subset \mathbb{V}$ is available as the labeled node set and usually we have $|\mathbb{V}^L| \ll |\mathbb{V}|$. Based on those labeled nodes, a hypothesis model $f$ is trained to learn the unknown labeling function and to predict the class for unlabeled nodes. The vanilla cross-entropy loss is usually adopted to train the model, which is given as follows:

$$\min_f \mathcal{L}_{ce} = - \sum_{v \in \mathbb{V}^L} \sum_{y=1}^{R(Y)} \mathbf{1}(y_v == y) \cdot \log(p(\hat{y}_v == y)), \quad (1)$$

where $y_v$ is the ground-truth label of node $v$, $\mathbf{1}(y_v == y)$ indicates correctness of label $y$, $\hat{y}_v$ is the label predicted by model $f$, and $p()$ is the predicted probability.

### 3.2 Graph Neural Networks

Graph neural networks are able to learn from non-Euclidean data, combining relational signal processing and convolution kernels on graphs, and have shown improved empirical performance across a wide range of graph-based learning tasks [13, 60]. As shown in [14], most existing GNN layers can be summarized in the following message-passing framework:

$$\boldsymbol{m}_v^{l+1} = \sum_{u \in \mathcal{N}_v} \mathbf{M}_l(\boldsymbol{h}_v^l, \boldsymbol{h}_u^l, \boldsymbol{A}_{v,u}), \quad \boldsymbol{h}_v^{l+1} = \mathbf{U}_l(\boldsymbol{h}_v^l, \boldsymbol{m}_v^{l+1}) \quad (2)$$

where $\mathcal{N}_v$ is the set of neighbors of $v$ in $\mathbf{G}$ and $\boldsymbol{h}_v^l$ denotes representation of $v$ in the $l$-th GNN layer. $\boldsymbol{A}_{v,u}$ represents the edge between

$v$ and $u$. $\mathbf{M}_l$ and $\mathbf{U}_l$ are the message function and update function at layer $l$, respectively.

### 3.3 Ambiguity of GNNs

Despite their popularity, an increasing number of studies suggest that GNNs can result in ambiguous node representations due to the aggregation of neighborhood messages, particularly in scenarios involving noisy edges or heterophily graphs [32, 54]. In this section, we focus on the heterophily as an example, as noisy edges can also be perceived as resulting in nodes with high heterophily.

*Heterophily.* This problem arises in situations where connected nodes may possess different attributes or labels that are inconsistent among neighboring nodes, leading to a graph exhibiting low homophily [32, 66]. For example, the node-level homophily metric can be defined as:

$$H_{\text{node}}(\mathcal{G}) = \frac{1}{|\mathcal{V}|} \sum_{v \in \mathcal{V}} H_{\text{node}}^v = \frac{1}{|\mathcal{V}|} \sum_{v \in \mathcal{V}} \frac{\left| \{ u \mid u \in \mathcal{N}_v, Z_{u,:} = Z_{v,:} \} \right|}{d_v}, \quad (3)$$

where a low value indicates the existence of strong heterophily. Typically, $Z$ denotes node labels (for class-wise homophily) or node attribute groups (for attribute-level homophily), and a low $H_{\text{node}}(\mathcal{G})$ may result in nodes to have mixed representations.

The heterophily problem has been observed to degrade GNN performance significantly and many solutions have been proposed by designing more expressive GNN layers [3, 18] or refining graph structures [30, 34] for graphs with high heterophily. In this work, we argue that the inductive bias of message-passing exist across all graphs and may behave differently in various regions of the graph, and focus on identifying and disambiguating GNN models by providing richer optimization guidance.

## 4 ANALYZING GNN BEHAVIORS

To investigate the problem of ambiguity for GNNs on graphs and evaluate the influence of message propagation across different graph regions, we condut an in-depth examination of the behavior of GCN [24] on two real-world graphs: Computer [42] and Blog-Catalog [43]. The model is trained on the semi-supervised node classification task, with configurations following Section 6.1. After the model converges, we test its performance on different graph regions for a comprehensive understanding of its behavior. Next, we will first introduce two graph split strategies implemented for our analysis, followed by a discussion of our empirical findings.

### 4.1 Graph-split Strategies

Our aim is to understand how the message-passing mechanism influences model performance across regions that exhibit varying ego-graph patterns, such as regions at the boundary of different classes or those adjacent to minority nodes. To this end, we propose two graph-split strategies.

The *first strategy* is designed to analyze the relationship between GNN's accuracy, node labels, and the level of heterophily. Concretely, we start by grouping node classes into three categories based on their frequencies, namely Majority, Middle, and Minority classes. The frequency thresholds of three groups are obtained by evenly splitting the range of class frequencies into three pieces.

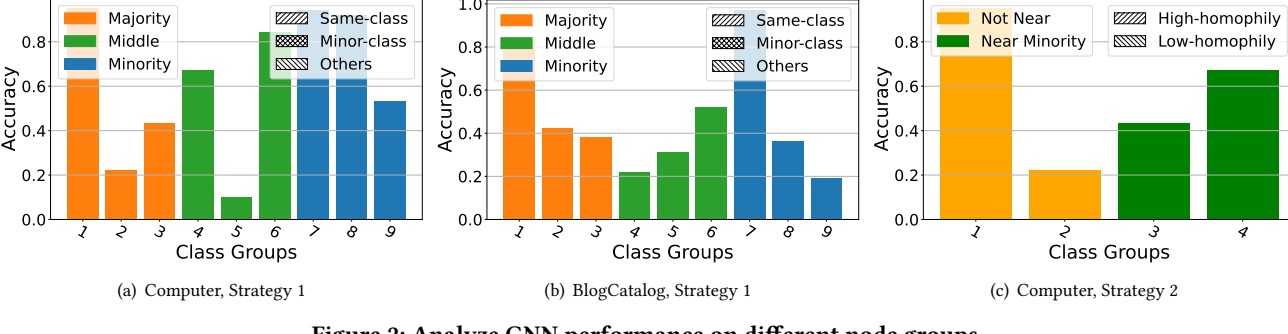

(a) Computer, Strategy 1      (b) BlogCatalog, Strategy 1      (c) Computer, Strategy 2

**Figure 2: Analyze GNN performance on different node groups.**

Nodes within each class are then further divided into three sub-groups based on their neighbors: those with high homophily (Same-class), those with high heterophily and most neighbors from minority classes (Minor-class), and those with high heterophily but most neighbors are not of minority classes (Others). The separation of heterophilous nodes into Minor and Others allows us to examine the influence of heterophily when most neighbors are from classes of different sizes.

The *second strategy* focuses specifically on nodes adjacent to minority classes. Initially, nodes are clustered into two groups based on whether they are connected to nodes of minority classes (minority classes are obtained in the same way as the last strategy). Subsequently, these groups are further divided based on the degree of homophily exhibited. This separation allows us to examine the performance of GNN at different positions, whether near the boundary of minority classes, and simultaneously evaluate the impact of heterophily.

### 4.2 Empirical Findings

In Fig. 2, we illustrate the accuracy of the trained GNN w.r.t each node group, under both splitting strategies. Several key observations can be derived from the results:

- The heterophily problem impacts nodes of different classes in diverse ways. For nodes belonging to majority classes, an increase in heterophily leads to a substantial drop in performance. However, this is not always the case for minority nodes (as seen in Fig.2(a)) and middle-sized nodes (as shown in Fig.2(b)). For these classes, higher heterophily does not necessarily equate to a decline in performance; in some instances, it may even enhance it.
- For nodes with high heterophily, the class of most neighbors would also influence GNN's performance (as shown in Fig.2(a)). For heterophilous nodes of majority and middle classes, when most neighbors are from minority classes, their performance drop would be larger.
- The detrimental effect of heterophily is considerably more significant in regions distant from minority classes than those in the proximity (as indicated in Fig. 2(c)).

These findings suggest that heterophily itself may not be the root of the issue, but rather the difficulty of learning a generalizable propagation mechanism. The varying impacts of heterophily could be attributed to difference in frequency of ego-graph patterns

across classes. For nodes within majority classes, a high-homophily ego-graph pattern may predominate, whereas, for nodes in minority classes, a substantial proportion may exhibit high heterophily, resulting in increased resilience against the effects of heterophily. Furthermore, during training, the pattern where most neighbors belong to minority classes might be relatively rare, which can lead to confusion in the aggregated message for nodes in these regions.

These observations inspire us to investigate the potential for enhancing the disambiguation of GNNs. This might be achieved by developing additional learning signals for nodes in ambiguous regions, rather than relying exclusively on the sparse labels available.

## 5 DISAMBIGUATED NODE REPRESENTATION LEARNING

To address the issue of ambiguity in node representations, which arises due to their positions in the graph and ego-graph structures, we introduce a novel framework called DisamGCL. This framework dynamically identifies ambiguous regions of the graph and guides their learning processes with contrasts in the neighborhood. DisamGCL can be easily utilized as an auxiliary objective for downstream tasks, and we will present details of it in this section.

### 5.1 Discovery of Nodes in Mixed Regions

The analysis in Sec. 4 indicates that nodes located in certain regions of graph, such as those where classes are mixed or exhibit minority neighborhood patterns, are subject to ambiguity due to GNN message-propagation. As GNNs are learned in a data-driven manner, the existence of these ambiguous regions may be sensitive to factors like the distribution of labeled data, specific GNN layer architectures, optimization strategies, etc. Therefore, one major challenge is efficiently identifying nodes with ambiguous representations. Prior studies have demonstrated that the model tend to exhibit unstable predictions across different training stages for instances that the model is difficult to learn or generalize [65]. Hence, we propose a method to identify nodes in ambiguous regions by analyzing the consistency of label prediction outputs.

Concretely, for each node $v$, we adopt a memory cell $\hat{\boldsymbol{e}}_v \in \mathbb{R}^{R(Y)}$ to encode the historical variance of predicted label distributions for

$v$. After each training epoch $t$, we update $\hat{e}_v$ as follows:

$$\hat{e}_v^t = \mu \cdot \hat{e}_v^{t-1} + (1 - \mu) \cdot p^t(\hat{y}_v),$$

$$s_v^t = \sum_{y=1}^{R(Y)} -\hat{e}_{v,y}^t \cdot \log(\hat{e}_{v,y}^t), \qquad (4)$$

where $\mu$ is a hyper-parameter determining the weight given to historical memory, and $p^t(\hat{y}_v)$ is the probability distribution over classes for node $v$ at time $t$, predicted by the current model. $\hat{e}_v$ encodes the historical prediction distribution of node $v$, and will be more flat for ambiguous nodes (less consistent predictions) while sharper for others (consistent predictions). Ambiguity score $s_v^t$ encodes the uncertainty and temporal variance for the prediction of node $v$, and is normalized to the scale $[0, 1]$. Nodes receiving mixed messages can be exposed with a high ambiguity score, and we select them using a threshold in the experiments.

## 5.2 Disambiguation with Augmented Contrasts

Based on previous analysis, one important reason of ambiguity is the inductive bias in message propagation. With noisy neighborhoods, aggregated messages may also be indiscriminative, resulting in ambiguous node representations with mixed semantics. To address this problem, we propose to disambiguate these nodes by augmenting the learning process with a contrastive learning objective, encouraging stronger distinctions from their dissimilar neighbors. Through contrasts in the embedding space, their representations will be guided to cluster towards closer node groups instead of staying in areas with mixed semantics.

Concretely, we use JSD loss [19] for contrastivity. For a selected ambiguous node $v$, the JSD contrastive objective is as follows:

$$\min_f \mathcal{L}_{cs,v} = \frac{1}{|\mathbb{Z}_v^+|} \sum_{z_u \in \mathbb{Z}_v^+} sp(-T(z_v, z_u)) + \frac{1}{|\mathbb{Z}_v^-|} \sum_{z_u \in \mathbb{Z}_v^-} sp(T(z_v, z_u)),$$

$$(5)$$

where $T(\cdot)$ represents a compatibility estimation function implemented as a dot-product, $f$ is the model for representation learning, and $sp(\cdot)$ denotes the *softplus* activation function. The positive and negative node groups $\mathbb{Z}_v^+$ and $\mathbb{Z}_v^-$ are selected based on semantic differences within the neighborhood to encourage distinction from dissimilar neighbors, which will be introduced below. The full contrastive loss is $\mathcal{L}_{cs} = \sum_{v \in \mathcal{V}'} \mathcal{L}_{cs,v}$, with $\mathcal{V}'$ denoting the set of ambiguous nodes.

As we focus on ambiguity that stems from message passing, we select $\mathbb{Z}_v^+$ and $\mathbb{Z}_v^-$ for $v$ based on its consistency with the neighborhood. Neighbors that carry higher semantic similarities should be selected as positive samples $\mathbb{Z}_v^+$ while those dissimilar ones should be included in $\mathbb{Z}_v^-$. This contrast will push $z_v$ closer to those of $\mathbb{Z}_v^+$ instead of demonstrating mixed semantics, thereby encouraging a more discriminative representation. Concretely, we utilize its neighbors as follows:

$$\mathbb{Z}_v^{t,+} := \{z_k \in N(z_v) \mid T(z_k, z_v) > \epsilon_1 \cdot \max_{z_u \in N(z_v)} T(z_u, z_v)\}$$

$$\mathbb{Z}_v^{t,-} := \{z_k \in N(z_v) \mid T(z_k, z_v) \leq \epsilon_2 \cdot \max_{z_u \in N(z_v)} T(z_u, z_v)\} \qquad (6)$$

$N(z_v)$ denotes the embeddings of nodes neighboring to target node $i$, and $\epsilon_1, \epsilon_2$ are controlling variables which are set to 0.75 and 0.4 respectively in experiments. Due to the problem of sparsity

---

**Algorithm 1** Disambiguated GNN Learning

**Require:** $G = (\mathbb{V}, \mathcal{E}; F, A)$, $\{y_v \mid v \in \mathbb{V}^L\}$
1: Random initialize model parameters
2: **for** $t$ in $t_{max}$ And Not Converged **do**
3:    **if** $\hat{e}, s$ is initialized **then**
4:       Select ambiguous nodes with a threshold on the normalized ambiguity score $s$
5:       For each ambiguous node, put its neighbors into positive and negative groups as Eq. 6
6:       For each ambiguous node, augment its positive group with similar distant nodes as Sec. 5.2
7:       Optimize $f$ to minimize $\mathcal{L}_{ce} + \lambda \mathcal{L}_{cs}$
8:    **else**
9:       Optimize $f$ to minimize $\mathcal{L}_{ce}$
10:    **end if**
11:    **if** $t\%T = 0$ **then**
12:       **if** $\hat{e}, s$ is initialized **then**
13:          Update historical prediction memory $\hat{e}$ and $s$ following Eq. 4
14:       **else**
15:          Initialize $\hat{e}$ with the predicted label prediction, compute $s$ as Eq. 4
16:       **end if**
17:    **end if**
18: **end for**
19: **return** Trained hypothesis model $f$

---

and potential heterophily issue, using neighbors alone may be insufficient for finding semantically consistent nodes and guiding the contrastive process. Addressing this issue, we further introduce embeddings of similar yet non-connected nodes: $\{z_u \notin N(z_v) \mid T(z_k, z_v) \geq \mathcal{T}\}$, in which $\mathcal{T}$ is the threshold of similarity. At each step we randomly sample $K$ instances from this auxiliary set to augment $\mathbb{Z}_v^{t,+}$. In experiments, $\mathcal{T}$ is set to 0.7 and $K$ is set to 8.

## 5.3 Overall Algorithm

The proposed $\mathcal{L}_{cs}$ can be seamlessly incorporated with the downstream node classification loss $\mathcal{L}_{ce}$ in Eq. 1. The full objective is given as:

$$\min_f \mathcal{L}_{ce} + \lambda \mathcal{L}_{cs}, \qquad (7)$$

where $\lambda$ controls the weight of our disambiguation loss. A detailed algorithm is summarized in Alg. 1. We will update the estimation of ambiguous nodes every $T$ iterations, and within each iteration, contrasts between identified nodes and their augmented neighbors will be conducted.

The proposed algorithm can also be used in together with other contrastive learning methods, which are designed to utilize those vast amount of unlabeled nodes.

## 6 EXPERIMENTS

We now demonstrate the effectiveness of DisamGCL in alleviating ambiguity of nodes in the mixed regions, through experiments on five node classification datasets. Particularly, we want to answer

Table 1: Results in node classification on three benchmark datasets.

| Method | Cora | | | BlogCatalog | | | Computer | | |
|---|---|---|---|---|---|---|---|---|---|
| | ACC | MacroF | AUROC | ACC | MacroF | AUROC | ACC | MacroF | AUROC |
| SRGNN | $80.5_{\pm1.5}$ | $79.6_{\pm1.7}$ | $89.3_{\pm0.9}$ | $45.7_{\pm2.2}$ | $45.1_{\pm1.6}$ | $82.0_{\pm1.4}$ | $76.2_{\pm1.3}$ | $75.8_{\pm2.3}$ | $86.8_{\pm0.7}$ |
| DropEdge | $81.1_{\pm1.5}$ | $80.2_{\pm1.1}$ | $89.6_{\pm0.7}$ | $45.5_{\pm1.8}$ | $45.2_{\pm1.6}$ | $82.2_{\pm1.2}$ | $75.9_{\pm1.7}$ | $75.9_{\pm2.6}$ | $86.9_{\pm0.6}$ |
| Focal | $80.9_{\pm1.3}$ | $79.8_{\pm1.6}$ | $89.6_{\pm0.7}$ | $45.2_{\pm2.1}$ | $44.6_{\pm1.7}$ | $81.7_{\pm1.5}$ | $77.3_{\pm1.9}$ | $75.7_{\pm2.1}$ | $87.0_{\pm0.5}$ |
| ReNode | $81.4_{\pm1.6}$ | $80.6_{\pm1.8}$ | $89.7_{\pm0.6}$ | $45.9_{\pm2.3}$ | $45.6_{\pm1.5}$ | $82.7_{\pm1.3}$ | $77.1_{\pm1.5}$ | $75.6_{\pm2.1}$ | $86.6_{\pm0.7}$ |
| TopoImb | $80.6_{\pm1.7}$ | $79.3_{\pm1.9}$ | $89.5_{\pm1.1}$ | $46.1_{\pm1.8}$ | $45.6_{\pm1.9}$ | $82.1_{\pm1.1}$ | $77.4_{\pm1.9}$ | $75.9_{\pm1.9}$ | $87.1_{\pm0.5}$ |
| CE | $80.8_{\pm1.2}$ | $79.9_{\pm1.5}$ | $89.8_{\pm0.8}$ | $45.3_{\pm2.1}$ | $44.8_{\pm1.8}$ | $81.9_{\pm1.3}$ | $76.9_{\pm1.6}$ | $75.5_{\pm2.4}$ | $86.7_{\pm0.6}$ |
| + DisamGCL | $81.5_{\pm1.3}$ | $\mathbf{81.0_{\pm1.4}}$ | $89.7_{\pm0.6}$ | $47.2_{\pm2.2}$ | $47.6_{\pm1.3}$ | $\mathbf{85.3_{\pm0.5}}$ | $78.3_{\pm1.5}$ | $76.3_{\pm2.3}$ | $87.4_{\pm0.4}$ |
| SupCon | $80.6_{\pm1.5}$ | $79.7_{\pm1.7}$ | $89.7_{\pm0.9}$ | $46.5_{\pm2.3}$ | $46.2_{\pm1.8}$ | $82.5_{\pm1.6}$ | $78.7_{\pm1.3}$ | $77.1_{\pm1.9}$ | $88.7_{\pm0.7}$ |
| + DisamGCL | $\mathbf{81.7_{\pm1.4}}$ | $80.9_{\pm1.6}$ | $89.8_{\pm0.5}$ | $\mathbf{48.6_{\pm1.9}}$ | $48.5_{\pm1.2}$ | $84.9_{\pm0.6}$ | $\mathbf{80.1_{\pm1.7}}$ | $79.6_{\pm2.5}$ | $\mathbf{89.3_{\pm0.5}}$ |
| DGI | $80.7_{\pm1.9}$ | $79.6_{\pm2.0}$ | $89.8_{\pm0.8}$ | $45.9_{\pm1.8}$ | $46.4_{\pm2.2}$ | $82.7_{\pm1.3}$ | $78.8_{\pm1.4}$ | $74.7_{\pm2.4}$ | $88.8_{\pm0.6}$ |
| + DisamGCL | $81.3_{\pm1.2}$ | $80.3_{\pm1.5}$ | $89.7_{\pm0.5}$ | $48.3_{\pm1.6}$ | $\mathbf{48.6_{\pm1.6}}$ | $85.1_{\pm0.8}$ | $79.6_{\pm1.8}$ | $76.5_{\pm2.2}$ | $89.1_{\pm0.8}$ |

Table 2: Results in node classification on three heterophily graph datasets.

| Method | Squirrel | | | Chameleon | | | Actor | | |
|---|---|---|---|---|---|---|---|---|---|
| | ACC | MacroF | AUROC | ACC | MacroF | AUROC | ACC | MacroF | AUROC |
| SRGNN | $55.9_{\pm1.4}$ | $56.1_{\pm1.8}$ | $78.2_{\pm0.8}$ | $54.1_{\pm1.9}$ | $55.6_{\pm1.6}$ | $83.1_{\pm0.8}$ | $50.5_{\pm1.8}$ | $48.1_{\pm1.3}$ | $85.4_{\pm0.9}$ |
| DropEdge | $57.3_{\pm1.6}$ | $56.7_{\pm1.7}$ | $78.7_{\pm0.9}$ | $53.7_{\pm2.1}$ | $55.3_{\pm1.5}$ | $83.0_{\pm0.8}$ | $51.3_{\pm1.1}$ | $48.9_{\pm1.2}$ | $86.3_{\pm0.8}$ |
| Focal | $54.7_{\pm2.1}$ | $55.2_{\pm2.3}$ | $78.1_{\pm0.9}$ | $53.3_{\pm2.5}$ | $54.7_{\pm1.7}$ | $82.4_{\pm1.3}$ | $50.9_{\pm1.2}$ | $48.7_{\pm1.4}$ | $85.6_{\pm0.6}$ |
| ReNode | $56.3_{\pm1.8}$ | $56.6_{\pm1.9}$ | $78.8_{\pm1.0}$ | $54.3_{\pm1.7}$ | $55.7_{\pm1.7}$ | $83.1_{\pm0.9}$ | $51.4_{\pm1.8}$ | $48.8_{\pm1.6}$ | $85.9_{\pm1.1}$ |
| TopoImb | $57.1_{\pm1.5}$ | $56.8_{\pm1.5}$ | $79.1_{\pm0.3}$ | $54.5_{\pm2.2}$ | $56.1_{\pm1.4}$ | $82.8_{\pm0.8}$ | $51.3_{\pm1.5}$ | $48.3_{\pm1.5}$ | $85.9_{\pm0.8}$ |
| CE | $56.8_{\pm1.6}$ | $56.3_{\pm1.6}$ | $78.4_{\pm0.6}$ | $53.6_{\pm2.3}$ | $55.3_{\pm1.7}$ | $82.9_{\pm0.9}$ | $50.8_{\pm1.4}$ | $48.5_{\pm1.2}$ | $85.6_{\pm0.7}$ |
| + DisamGCL | $\mathbf{57.6_{\pm1.7}}$ | $\mathbf{57.2_{\pm1.4}}$ | $79.5_{\pm0.7}$ | $55.6_{\pm1.9}$ | $56.4_{\pm1.4}$ | $83.2_{\pm0.4}$ | $51.9_{\pm1.1}$ | $49.1_{\pm2.5}$ | $87.7_{\pm0.4}$ |
| SupCon | $55.4_{\pm1.9}$ | $55.2_{\pm1.7}$ | $77.8_{\pm0.5}$ | $55.7_{\pm2.1}$ | $55.9_{\pm1.8}$ | $83.2_{\pm0.7}$ | $51.2_{\pm2.5}$ | $49.7_{\pm1.2}$ | $86.7_{\pm0.8}$ |
| + DisamGCL | $56.9_{\pm1.9}$ | $57.1_{\pm1.6}$ | $78.4_{\pm0.8}$ | $56.6_{\pm1.8}$ | $56.9_{\pm1.6}$ | $\mathbf{83.5_{\pm0.5}}$ | $53.2_{\pm1.6}$ | $\mathbf{52.3_{\pm1.9}}$ | $\mathbf{88.4_{\pm0.5}}$ |
| DGI | $53.9_{\pm2.1}$ | $52.7_{\pm1.5}$ | $77.6_{\pm0.5}$ | $53.5_{\pm2.2}$ | $54.4_{\pm2.1}$ | $82.3_{\pm0.7}$ | $51.9_{\pm1.7}$ | $49.1_{\pm1.4}$ | $86.4_{\pm0.9}$ |
| + DisamGCL | $55.7_{\pm1.6}$ | $56.1_{\pm1.8}$ | $\mathbf{80.3_{\pm0.6}}$ | $\mathbf{56.7_{\pm2.3}}$ | $\mathbf{56.9_{\pm1.6}}$ | $83.4_{\pm0.5}$ | $\mathbf{53.4_{\pm1.6}}$ | $51.2_{\pm2.6}$ | $88.1_{\pm0.6}$ |

the following research questions: (i) **RQ1** Can the proposed DisamGCL improve the overall performance of node classification? (ii) **RQ2** Would DisamGCL successfully detect ambiguous nodes in the mixed regions? (iii) **RQ3** How would DisamGCL generalize to different GNN layer variants? And how sensitive would DisamGCL be towards its weight and the threshold of ambiguity score?

## 6.1 Experiment Settings

**Datasets.** In experiment, we adopt 6 real-world datasets, including three benchmarks with low heterophily: *Cora* [41], *BlogCatalog* [43] and *Computer* [42], and three benchmarks frequently used as graphs exhibiting varying degrees of heterophily: *Squirrel* [66], *Chameleon* [66] and *Actors* [34]. Details of these datasets are provided below.

- **Cora** The Cora dataset is a citation network used for transductive node classification. It comprises a single large graph with 2, 708 nodes representing academic papers across 7 different fields (classes). Each node attribute is derived from a *bag-of-words* representation of the paper's content, and the graph includes a total of 5, 429 citation edges.

- **BlogCatalog** The BlogCatalog dataset[1] is a social network containing 10, 312 nodes (bloggers) from 38 classes and 333, 983 friendship edges. Each node is associated with a 64-dimensional embedding vector obtained using DeepWalk, as described in [35].

- **Computer** This Amazon product co-purchase network features 13, 752 nodes representing products across 10 categories (classes), and includes 491, 722 edges. Each edge denotes that the corresponding products are frequently bought together. The attributes for each node are derived from *bag-of-words* representations of the product reviews.

- **Squirrel, Chameleon** The Squirrel and Chameleon datasets [34] consist of Wikipedia web pages discussing specific topics. They are frequently used as examples of graphs exhibiting varying degrees of heterophily [66]. Nodes represent web pages and edges denote mutual links. Nodes are categorized into 5 classes. The Squirrel dataset contains 5, 201 nodes and 401, 907 edges, whereas Chameleon comprises 2, 277 nodes and 65, 019 edges.

- **Actor** The Actor dataset is a social network describing relationships among a set of actors (nodes), and is often utilized as a benchmark for graphs with high heterophily. Both node attributes and edges are extracted from Wikipedia descriptions,

[1]http://www.blogcatalog.com

and the task is to categorize actors into five different classes. The dataset includes $4,600$ nodes and $30,019$ edges in total.

**Baselines.** To conduct a comprehensive empirical comparison, we consider two categories of baseline. (1) We test three learning frameworks, including the conventional node classification based on cross-entropy (CE) and two contrastive learning methods Sup-Con [69] and DGI [45]. Note that DGI is originally designed for unsupervised learning. We implement it in the joint-learning setting for fairer comparisons, the same as others. (2) We compare with several augmented learning strategies that can be used for GNNs. Specifically, SRGNN [68] incorporates distribution of test nodes into training, Focal loss [29] emphasizes those challenging-to-learn nodes, DropEdge [38] augments the graph by creating more diverse neighborhood patterns, ReNode [5] reweights labeled nodes based on their relative positions and TopoImb [59] considers the frequency of ego-graph structures of labeled nodes.

**Configurations.** All experiments are conducted on a 64-bit machine with Nvidia A6000, and ADAM optimization algorithm is used to train all the models. Learning rate is initialized to 0.001, with weight decay being $5e-4$ and the maximum training epoch as $8,000$. For all datasets, the train/validation/test split is set to $0.5 : 1 : 8.5$. If not emphasized otherwise, a two layer GCN is adopted, $\lambda$ is set to 1.0, and threshold for selecting ambiguous nodes is set to 0.8.

**Evaluation Metrics.** Following existing works [23, 39], we adopt three criteria: classification accuracy(ACC), Macro F-measure, and mean AUCROC score. ACC is computed on all testing examples at once, AUC-ROC score illustrates the probability that the corrected class is ranked higher than other classes, and Macro F gives the harmonic mean of precision and recall for each class. Both AUCROC score and Macro F are calculated separately for each class and then non-weighted average over them, therefore can better reflect the performance on minority groups.

## 6.2 DisamGCL for Node Classification

To answer **RQ1**, in this section, we compare the performance on node classification between proposed DisamGCL and all aforementioned baselines. DisamGCL is incorporated into all three learning frameworks, CE, SupCon and DGI. Models are tested on 6 real-world datasets, and each experiment is conducted 3 times to alleviate the randomness. The average results with standard deviation for three datasets with low heterophily are reported in Table 1 and those for three datasets with high heterophily are reported in Table 2.

From the tables, we can observe that our proposed DisamGCL shows a consistent improvement across all three learning frameworks, outperforming all baselines on six datasets with a clear margin. For example, DisamGCL shows an improvement of 2.2 point in Macro F on BlogCatalog and 2.5 point in Macro F on chameleon for the DGI backbone. These results indicate that through automatic discovery of ambiguous nodes and neighborhood-aware contrasts, our approach is better at learning representations for nodes. Besides, the improvement tends to be larger in terms of Macro F score compared to mean accuracy, which indicates that DisamGCL is beneficial for minority classes.

## 6.3 Ability of Detecting Nodes in Mixed Regions

To address **RQ2**, we visualize the average ambiguity scores assigned to node groups defined in Section 4. We present the results of the variant *CE+DisamGCL* in Fig. 3. In this figure, the blue bars represent the average accuracy within each node group, while the green bars indicate the average ambiguity score.

From the visualization, it is evident that for both Computer and BlogCatalog datasets, groups with lower accuracy tend to be assigned higher ambiguity scores. Notably, nodes belonging to minority classes, exhibiting high heterophily, and located near class boundaries are typically assigned larger weights. These findings validate the ability of our proposed DisamGCL in accurately identifying nodes with greater ambiguity in their classification.

## 6.4 DisamGCL for Different GNN Backbones

To answer **RQ3** w.r.t generality across GNN variants,we vary the GNN backbone across GCN [24], Sage [15], GIN [53], and SGC [49]. We examine the performance of the CE both before and after incorporating DisamGCL. Given the changed model architecture, the weight $\lambda$ is adjusted through a grid search to achieve optimal performance, while all other configurations remain unchanged as Sec. 6.1. Each experiment is randomly run for 3 times on Cora, BlogCatalog and Actor, with results in Tab. 3. The observed results demonstrate a consistent performance improvement across all settings, validating both the generality and efficacy of our proposed method across diverse GNN architectures.

## 6.5 Sensitivity Analysis

To answer **RQ3** concerning hyperparameter sensitivity, we conduct a set of analysis with model *CE+DisamGCL*. Unless specified otherwise, all configurations remain unchanged, and each experiment is randomly run 3 times. Results are presented below.

**Hyperparameter $\lambda$.** Weight of our proposed contrastive loss in Eq. 7 is analyzed in Fig. 4. It can be observed that DisamGCL performs relatively better with $\lambda$ set to the range $[0.8, 1.0]$. A performance decline can be observed when $\lambda$ exceeds 1.0, as $\mathcal{L}_{cs}$ could be noisy and overshadow node classification loss.

**Ambiguity Threshold.** As shown in Fig. 5, a threshold value around $[0.6, 0.8]$ yields the best results. A threshold that is too low could lead to the identification of non-ambiguous nodes (as Fig. 3), which could undermine the effectiveness of the neighborhood-wise contrasts. On the other hand, a too high threshold will fail to detect nodes in ambiguous regions.

**Ambiguity Estimation.** We further analyze the hyperparameter for ambiguity estimation, $\mu$ in Eq. 4. From Fig. 6, it can be observed that setting it within $[0.5, 0.6]$ may obtain the better performance, and the sensitivity towards it is low on both Computer and Actor datasets.

## 7 CONCLUSION

In this study, we delved into the challenge of performance degradation experienced by GNNs in specific graph regions, namely 'ambiguous regions'. We identified that this degradation arises from the inherent inductive bias of message propagation and the under-representation of such regions during the training process. A novel framework, Method Name, is designed to autonomously detect

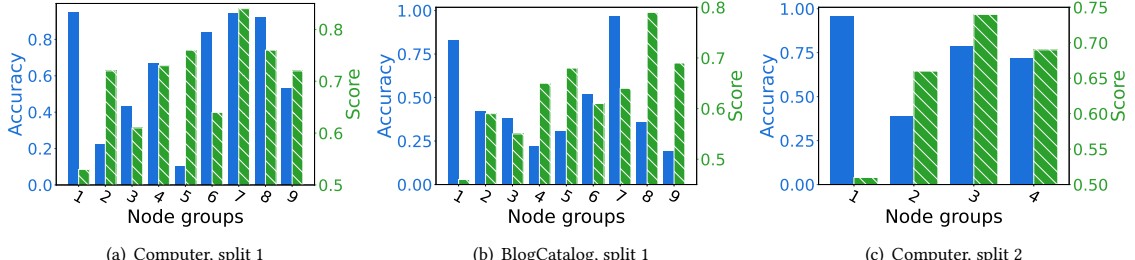

Figure 3: Analyze ambiguity score on different node groups.

Table 3: Results of DisamGCL in node classification with varying GNN backbones.

| Model | Cora | | | BlogCatalog | | | Actor | | |
|---|---|---|---|---|---|---|---|---|---|
| | ACC | MacroF | AUROC | ACC | MacroF | AUROC | ACC | MacroF | AUROC |
| Sage | $83.7_{\pm1.3}$ | $81.8_{\pm1.7}$ | $89.8_{\pm0.7}$ | $46.4_{\pm1.7}$ | $46.2_{\pm1.7}$ | $82.1_{\pm1.5}$ | $73.8_{\pm1.2}$ | $69.7_{\pm1.5}$ | $86.4_{\pm0.8}$ |
| +DisamGCL | $84.9_{\pm1.4}$ | $82.4_{\pm1.4}$ | $89.9_{\pm0.8}$ | $47.6_{\pm1.4}$ | $47.8_{\pm1.8}$ | $84.7_{\pm1.9}$ | $74.6_{\pm1.3}$ | $70.8_{\pm1.6}$ | $88.3_{\pm0.9}$ |
| GCN | $80.8_{\pm1.2}$ | $79.9_{\pm1.5}$ | $89.8_{\pm0.8}$ | $45.3_{\pm2.1}$ | $44.8_{\pm1.8}$ | $81.9_{\pm1.3}$ | $50.8_{\pm1.4}$ | $48.5_{\pm1.2}$ | $85.6_{\pm0.7}$ |
| +DisamGCL | $81.5_{\pm1.3}$ | $81.0_{\pm1.4}$ | $89.7_{\pm0.6}$ | $47.2_{\pm2.2}$ | $47.6_{\pm1.3}$ | $85.3_{\pm0.5}$ | $51.9_{\pm1.1}$ | $49.1_{\pm2.5}$ | $87.7_{\pm0.4}$ |
| GIN | $82.3_{\pm1.6}$ | $80.1_{\pm1.8}$ | $89.7_{\pm0.9}$ | $46.3_{\pm1.6}$ | $46.1_{\pm1.9}$ | $82.1_{\pm1.1}$ | $71.1_{\pm1.6}$ | $68.6_{\pm1.3}$ | $85.8_{\pm0.9}$ |
| +DisamGCL | $83.5_{\pm1.5}$ | $81.8_{\pm1.6}$ | $89.8_{\pm1.2}$ | $47.4_{\pm1.9}$ | $47.7_{\pm2.1}$ | $84.7_{\pm1.2}$ | $72.9_{\pm1.5}$ | $69.9_{\pm1.7}$ | $87.9_{\pm0.6}$ |
| SGC | $78.5_{\pm1.6}$ | $78.6_{\pm1.8}$ | $89.2_{\pm1.1}$ | $43.6_{\pm1.8}$ | $43.7_{\pm1.7}$ | $81.4_{\pm1.8}$ | $66.7_{\pm1.9}$ | $63.6_{\pm1.6}$ | $83.2_{\pm0.7}$ |
| +DisamGCL | $80.3_{\pm1.6}$ | $79.8_{\pm1.3}$ | $89.5_{\pm0.7}$ | $45.9_{\pm2.3}$ | $45.9_{\pm1.6}$ | $83.4_{\pm1.2}$ | $70.6_{\pm1.5}$ | $67.7_{\pm2.1}$ | $84.9_{\pm0.6}$ |

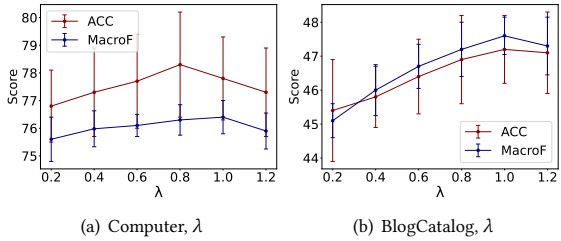

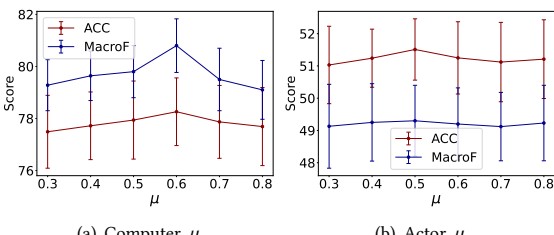

Figure 4: Sensitivity analysis over the weight of our proposed neighborhood-wise contrasts.

Figure 6: Sensitivity analysis over $\mu$ for updating node ambiguity estimation.

towards multimodal graphs: extending our approach to address ambiguity in graphs with multiple types of nodes, edges, or attributes. We also plan to integrate with other self-supervised learning strategies, which holds the potential to further boost GNNs' robustness and generalization capabilities.

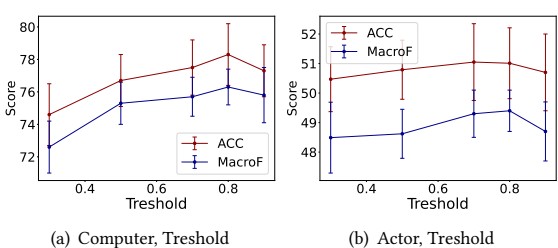

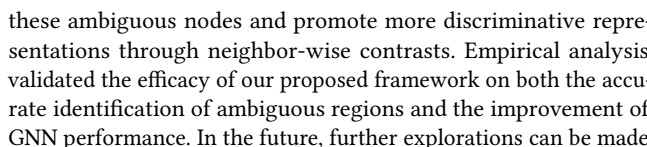

Figure 5: Sensitivity analysis over the threshold for identifying ambiguous nodes.

these ambiguous nodes and promote more discriminative representations through neighbor-wise contrasts. Empirical analysis validated the efficacy of our proposed framework on both the accurate identification of ambiguous regions and the improvement of GNN performance. In the future, further explorations can be made

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
