# OpenReview forum: "Disambiguated Node Classification with Graph Neural Networks"
_ACM.org/TheWebConf/2024/Conference — TheWebConf24_

### Official Review · Reviewer_MN9P · 2023-11-12

**Novelty:** 4
**Technical Quality:** 4

**Review:**

This work addresses ambiguity in Graph Neural Networks (GNNs) by introducing DisamGCL, a novel method that identifies and disambiguates nodes using temporal inconsistency and contrastive learning. DisamGCL enhances representation learning, particularly in ambiguous regions, improving GNN performance in underrepresented graph areas.

**Questions:**

- The author's starting point is commendable, with a strong motivation. However, the adopted detection and correction methods are relatively singular, relying solely on predictive consistency, which may be insufficient to truly capture minority regions.
- The experimental section of the paper is overly standardized, lacking more distinctive experimental explanations or visualizations to demonstrate the accuracy of detecting minority regions.
- What is the training time/cost comparison for your whole model and corresponding original model? It would be very helpful if the authors could provide a specific training time/cost compared with the baseline methods.
- The model is evaluated only on small datasets and doesn't know the scalability on large-scale datasets.
- Since latest baselines are from 2022, it is highly recommended for the authors to compare some recent methods (2023) to increase the persuasiveness of the results.
- As a research direction in machine learning and data mining on graphs, it is necessary to have some up-to-date surveys and related works. (i) A Comprehensive Survey on Deep Graph Representation Learning. 2023 (ii) Graph Neural Networks- Taxonomy, Advances and Trends. 2022 (iii) Deep Learning on Graphs: A Survey. 2022
- The author should provide a code repository link to ensure the replicability of the results.

**Reviewer Confidence:**

3: The reviewer is confident but not certain that the evaluation is correct

**Scope:**

4: The work is relevant to the Web and to the track, and is of broad interest to the community

---

### Official Review · Reviewer_XJvR · 2023-11-18

**Novelty:** 5
**Technical Quality:** 4

**Review:**

Strengths:
1. The authors conduct a motivation experiment to divide nodes into several categories for easier understanding.
2. The concept of ambiguous nodes was proposed, which reduces classification performance to a certain extent.
3. The effectiveness of ambiguous nodes was verified through experiments, and whether ambiguous nodes can be successfully detected was analyzed.

Weaknesses:
1. Section 4.2 states that heterophily is not the root cause of the problem, but the idea behinds reading the article is to aggregate ambiguous nodes closer to similar nodes during aggregation, which is also a method used to solve heterophily.
2.  Can we draw a graph to represent such a structure for ambiguous nodes?
3.  There is no explanation for using a memory cell or why it is possible to find ambiguous nodes.

**Questions:**

1. What is the initial value of the memory cell for finding ambiguous nodes?
2. Can the selection of neighboring nodes for ambiguous nodes in section 5.2 be understood as selecting a portion of useful nodes for aggregation for ambiguous nodes?
3. Why are the screening of similar and non similar nodes in section 3. 5.2 being 0.75 and 0.4? Will there be any overlap between these two?

**Reviewer Confidence:**

4: The reviewer is certain that the evaluation is correct and very familiar with the relevant literature

**Scope:**

3: The work is somewhat relevant to the Web and to the track, and is of narrow interest to a sub-community

---

### Official Review · Reviewer_9nR5 · 2023-11-19

**Novelty:** 5
**Technical Quality:** 5

**Review:**

In this paper, the authors study the ambiguity problem within GNNs, its impact on representation learning, and the development of richer supervision signals to fight against this problem. Based on this, authors conduct a fine-grained evaluation of GNN, analyzing the existence of ambiguity in different graph regions and its relation with node positions. The experimental results also illustrate the effectiveness of the authors' proposed method.

**Questions:**

1. Can the authors provide some theoretical analysis of section 4.2?

2. Why is JSD loss used within Line 493? Did the authors consider other loss functions such as KL loss and MSE loss?

3. In the experimental analysis (section 6.3-6.5), the author's analysis of the experimental results is not detailed enough.

4. What is the relevance of the inductive bias of message propagation in GNNs proposed by the authors and the over-smoothing problem in graph.

**Reviewer Confidence:**

3: The reviewer is confident but not certain that the evaluation is correct

**Scope:**

3: The work is somewhat relevant to the Web and to the track, and is of narrow interest to a sub-community

---

### Official Review · Reviewer_YgYi · 2023-11-26

**Novelty:** 5
**Technical Quality:** 4

**Review:**

This paper addresses the node classification problem on graphs with a mixture of homophily and heterophily. A characterization of the impact of homophily/heterophily for different node groups based on the popularity (majority/minority) of their classes motivates the design of DisamGCL. The proposed approach identifies so-called ambiguous nodes, which are those with less consistent predictions across classes. Predictions for such nodes are then disambiguated using contrastive learning where other similar nodes are considered as positive pairs. The proposed approach (combined with three different GNN backbones) is compared against multiple baselines in terms of node classification using datasets with both homophily and heterophily. The results show that DisamGCL often improves the performance of the base GNN.

Strengths:

+Results based on multiple datasets and evaluation metrics

+Multiple GNN backbones are considered in the experiments

+The proposed characterization seems to support the proposed method

Weaknesses:

-The paper is poorly written

-The gains of the proposed method are small

-The contributions of the proposed method are limited

Detailed comments:

This is a borderline paper with some potentially innovative ideas but poor experimental results and overall confusing writing. Here are some details on its weaknesses:

Writing: I only was able to understand what the paper is proposing after reading Section 5.2. The intro and even the characterization don’t do a great job of selling the paper's contributions. I am not familiar with the notion of ambiguity but uncertainty is something more commonly used in ML. The conclusions made in Section 4 are based on only two datasets, so it is unclear whether these are general patterns.

Performance gains: The gains reported in Tables 1 and 2 are small and it is not clear that the approaches considered are state-of-the-art for node classification (the baselines considered are tailored for specific settings, such as distribution shifts). The datasets considered are also quite small and do not include any of the OGBL datasets. Running times of the proposed method are also not provided.

Contributions: The key idea of the paper, which is applying contrastive learning selective for nodes with higher classification uncertainty alone doesn’t seem very novel. For instance, JGCL (not cited) seems to follow a similar approach:

Selahattin Akkas and Ariful Azad. 2022. JGCL: Joint Self-Supervised and Supervised Graph Contrastive Learning. In Companion Proceedings of the Web Conference 2022 (WWW '22). Association for Computing Machinery, New York, NY, USA, 1099–1105. https://doi.org/10.1145/3487553.3524722

**Questions:**

1) What is the significance of the gains presented in the experiments?

2) Why are these baselines selected? Are they state-of-the-art?

3) What is the running time of the proposed algorithm compared to the baselines?

4) Why are results for larger datasets missing?

5) What are the key contributions of the paper and how do they compare with the reference above (JGCL)?

**Reviewer Confidence:**

4: The reviewer is certain that the evaluation is correct and very familiar with the relevant literature

**Scope:**

4: The work is relevant to the Web and to the track, and is of broad interest to the community

---

### Official Review · Reviewer_2jDN · 2023-11-29

**Novelty:** 3
**Technical Quality:** 3

**Review:**

Summary:

The paper addresses a critical problem in the domain of graph-structured data analysis using Graph Neural Networks (GNNs). It focuses on the challenges posed by ambiguous regions within graphs, particularly those with varying degrees of heterophily and neighborhood patterns. These ambiguous regions often exhibit underrepresentation in training datasets, leading to performance degradation. The authors argue that the inductive bias inherent in GNNs' message-passing process contributes to this problem, as it tends to overlook such ambiguous regions, resulting in less effective node classification.


Pros:

(1) Interesting Motivation: The focus on disambiguating nodes in ambiguous regions by analyzing the stability of node representation learning is an intriguing motivation.

(2) Comprehensive Experiments: The paper conducts extensive experiments to validate its approach and framework, providing empirical evidence for the effectiveness of DisamGCL.

(3) Solution-Oriented: The introduction of a novel framework, DisamGCL, to tackle the identified problem is a significant step forward in enhancing GNN's capabilities in complex graph structures.

Cons:

(1) Unclear Scope: The relationship between minority nodes and heterophily is not clearly defined, possibly due to writing issues, leading to confusion about the exact scope and focus of the study.

(2) Less Exciting Findings: The major discoveries presented may seem too intuitive or obvious, lacking the excitement for the readers. For example, GNN shows the performance degradation in underrepresented graph regions containing different types of nodes. But I kind of doubt if the proposed method can solve this problem effectively since technically it's still designed for homophily of graphs.

(3) Incomplete Experimental Analysis: The absence of certain critical experiments, such as comparisons with heterophily methods and incomplete analysis of parameter sensitivity, leaves gaps in the validation of the proposed approach.

**Questions:**

(1) according to Sec 5.2, the paper selects the positive and negative samples for contrastive learning based on semantic differences to encourage distinction from dissimilar neighbors. However, this strategy seems a bit contradictory from the goal of handling potential heterophily issue.

(2) The analysis in Fig 2 is interesting. I expect to see if the results of blogcatalog with strategy 2 also follows the discovery mentioned in the paper.

(3) Any principle to choose the value for hyperparameter? like epsilon 1 and epsilon 2 in equation 6

**Ethics Review Description:**

no issue

**Reviewer Confidence:**

3: The reviewer is confident but not certain that the evaluation is correct

**Scope:**

4: The work is relevant to the Web and to the track, and is of broad interest to the community

---

### Decision · Program_Chairs · 2024-01-22

**Decision:**

Accept

**Comment:**

This paper focuses on the concept of ambiguity in Graph Neural Networks and is proposing a method which is specifically desinged to directly address ambiguous regions of the graph towards improving performance. The proposed method appears to work well compared to baselines.

 Overall, most significant reviewers concerns have to do with the experimental evaluation, many of which are directly addressed during the rebuttal, while some left unaddressed (such as the inclusion of heterophily baseline methods)

 Finally, there are some concerns regarding the quality and clarity of presentation, which should be a point of focus in revising the paper.